# Identification and Preliminary Clinical Validation of Key Extracellular Proteins as the Potential Biomarkers in Hashimoto’s Thyroiditis by Comprehensive Analysis

**DOI:** 10.3390/biomedicines11123127

**Published:** 2023-11-24

**Authors:** Zihan Xi, Tinglin Yang, Tao Huang, Jun Zhou, Peng Yang

**Affiliations:** Department of Breast and Thyroid Surgery, Union Hospital, Tongji Medical College, Huazhong University of Science and Technology, Wuhan 430022, China

**Keywords:** autoimmune thyroid disease, Hashimoto’s thyroiditis, extracellular protein, bioinformatics, chemokine

## Abstract

Hashimoto’s thyroiditis (HT) is an autoimmune disruption manifested by immune cell infiltration in thyroid tissue and the production of antibodies against thyroid-specific antigens, such as the thyroid peroxidase antibody (TPOAb) and thyroglobulin antibody (TGAb). TPOAb and TGAb are commonly used in clinical tests; however, handy indicators of the diagnosis and progression of HT are still scarce. Extracellular proteins are glycosylated and are likely to enter body fluids and become readily available and detectable biomarkers. Our research aimed to discover extracellular biomarkers and potential treatment targets associated with HT through integrated bioinformatics analysis and clinical sample validations. A total of 19 extracellular protein-differentially expressed genes (EP-DEGs) were screened by the GSE138198 dataset from the Gene Expression Omnibus (GEO) database and protein annotation databases. Gene Ontology (GO) and Kyoto Encyclopedia of Genes and Genomes (KEGG) were used to analyze the function and pathway of EP-DEGs. STRING, Cytoscape, MCODE, and Cytohubba were used to construct a protein–protein interaction (PPI) network and screen key EP-DEGs. Six key EP-DEGs (CCL5, GZMK, CXCL9, CXCL10, CXCL11, and CXCL13) were further validated in the GSE29315 dataset and the diagnostic curves were evaluated, which all showed high diagnostic accuracy (AUC > 0.95) for HT. Immune profiling revealed the correlation of the six key EP-DEGs and the pivotal immune cells in HT, such as CD8+ T cells, dendritic cells, and Th2 cells. Further, we also confirmed the key EP-DEGs in clinical thyroid samples. Our study may provide bioinformatics and clinical evidence for revealing the pathogenesis of HT and improving the potential diagnosis biomarkers and therapeutic strategies for HT.

## 1. Introduction

Hashimoto’s thyroiditis (HT), also called chronic lymphocytic thyroiditis, is the most frequent autoimmune thyroid disease characterized by the attack and destruction of the thyroid gland by the immune system [1]. HT can increase the risk of hypothyroidism and is associated with an increased risk of thyroid tumors, particularly papillary thyroid cancer (PTC) [2]. The incidence of HT is gradually increasing worldwide in the last 3 decades, with a prevalence of up to 5% [3,4]. HT has become one of the most common diseases of the thyroid, and around 2% of HT patients show clinical symptoms [5]. The incidence of HT also varies in different races, sexes, and ages. Higher incidence was more often found in the white race, women, or the elderly [6]. In addition, HT may also exhibit a higher prevalence in patients with autoimmune diseases including myasthenia gravis (MG), systemic sclerosis, etc. Despite further study being warranted focusing on the etiology and pathogenesis of HT, genetic susceptibility, environmental triggers, and autoimmune responses were closely relevant. The familial tendency is a significant feature of HT, indicating a genetic susceptibility to HT disease [7]. Genes that are coded in the human leukocyte antigen (HLA) complex are closely related to the development of HT, including the HLA-B* 46:01, HLA-A* 02:07, and HLA-DRB4 genes [8,9]. Previous studies also demonstrated that environmental triggers can prompt the development of HT together with innate drivers [10]. Dietary components including iodine, selenium, and vitamin D may play roles in HT. Excess iodine intake induces the onset of HT since highly iodinated thyroglobulin (Tg) can be more immunogenic. The lack of selenium reduces the protective role of selenium against the production of thyroid peroxidase antibody (TPOAb), and inverse relationships of vitamin D with TPOAb were also reported [11].

As a result of losing immune tolerance, the pathogenesis of HT is closely connected with autoantibodies and lymphocytic infiltration in thyroid tissue, including B cells and T cells [12]. In HT, autoreactive CD4+ T cells are activated, which recruit B cells and CD8+ T cells into the thyroid. Functionally alternated B cells can synthesize autoantibodies that attack thyroid cells. In addition, dysfunctional T cells aggravate the breakdown of immune homeostasis. Cytotoxic T lymphocytes that are specific to the thyroid also lead to autoimmune thyrocyte depletion [13,14]. To date, subjected to the atypical clinical presentations, the early diagnosis of HT is still insufficient. The TPOAb and thyroglobulin antibody (TGAb) are typical antibodies of the thyroid [3]. TPOAb is now considered the best serological diagnostic marker of HT, but there are still 5% of HT patients with this index negative. As for TGAb, it is positive in only 60–80% of HT patients, but positive in a greater proportion of healthy controls than TPOAb. The sensitivity of TPOAb is 90% for the diagnosis of HT, and TGAb has a sensitivity of 30–50% [15]. Therefore, it is essential for clinicians to seek novel biological markers of HT for studying the pathogenesis, improving diagnostic accuracy, and developing treatment strategies.

Since most nuclear proteins and cytoplasmic proteins cannot be detected in the body fluids of patients, we focused on finding extracellular biomarkers of HT in this study. Extracellular proteins are synthesized intracellularly and can be secreted into the extracellular environment, rendering them detectable in clinical tissues and certain body fluids. This characteristic attributes the potential value as diagnostic biomarkers or therapeutic targets for some diseases to extracellular proteins [16]. Recent studies have found that some molecules in the extracellular environment of thyroid tissue have immunomodulatory effects and are involved in the occurrence and development of HT [17], including a series of secreted cytokines and chemokines [18]. For instance, IL-1β, IL-6, IL-12, IL-13, IL-18, CCL21, CXCL13, CXCL12, and CCL22 have been reported to increase in thyroid tissues in patients with HT [19,20]. However, to our knowledge, no systematic study of extracellular proteins in HT has been reported yet.

Bioinformatics analysis of transcriptional profiling from microarray has emerged as a valuable tool for identifying disease-specific biomarkers, and several genes and pathways involved in HT metabolism and immunity have been identified [21,22,23]. In this study, we downloaded the normal and HT thyroid tissue microarray gene expression profile (GSE138198) from the Gene Expression Omnibus (GEO) database and screened out the differentially expressed genes (DEGs). Then, we screened the extracellular protein-differentially expressed genes (EP-DEGs) and performed biological function enrichment and pathway enrichment analysis. Key EP-DEGs were analyzed and validated in another GEO dataset GSE29315. Since HT is an autoimmune disease and different immune cells, including CD8+ T cells [24,25], dendritic cells (DCs) [18], and Th2 cells [26], have a significant influence on the development of HT, we performed immune infiltration analyses and valued the association of the key EP-DEGs with selected immune cells. Finally, clinical samples were collected to validate the expression of key EP-DEGs.

Despite the high incidence of HT, there are currently no therapeutic measures targeting the etiology and pathogenesis, and the mainstream treatment is based on the management of hypothyroidism with lifelong substitution therapy [27]. Our study aims to identify extracellular proteins that are involved in HT and find biomarkers that may improve the comprehension of pathogenic mechanisms, diagnostic protocols, and therapeutic strategies of HT clinically.

## 2. Materials and Methods

### 2.1. Study Design and Data Acquisition

The process of this study is shown in the flow chart (Figure 1). The microarray gene expression profile of the normal and HT thyroid tissues was acquired from the GEO database. Multiple analyses were performed using different packages from R (http://www.r-project.org, version 4.3.1, accessed date: 10 September 2023) in this study. DEGs were screened out in the dataset GSE138198, which contains 13 HT patients and 3 normal thyroid tissue samples and serves as the training set for initial screening and analysis. The extracellular protein gene lists were downloaded from the Uniprot database (accession code GO:0005576) [28] and the HPA protein annotation database [29], respectively. Further, EP-DEGs were screened out from DEGs and the extracellular protein gene lists. Subsequently, we used Gene Ontology (GO) and Kyoto Encyclopedia of Genes and Genomes (KEGG) databases to perform biological function enrichment and pathway enrichment analysis of EP-DEGs. Protein–protein interaction (PPI) network analyses of EP-DEGs were also established to screen out functional modules and hub EP-DEGs. To further validate the above findings, data from another dataset GSE29315 that contains 6 HT patients and 8 thyroid physiological hyperplasia (selected as controls) were applied to filter and identify the key EP-DEGs, followed by receiver operating characteristic (ROC) analysis of hub genes’ diagnostic accuracy on HT. The association of the key EP-DEGs with specific immune cells that are involved in the pathogenesis of HT was investigated by immune infiltration analysis. Finally, thyroid samples of normal and HT patients were collected to validate the expression of key EP-DEGs using RT-PCR, hematoxylin and eosin (H&E) staining, and immunohistochemistry (IHC) staining assays.

### 2.2. Identification of DEGs and EP-DEGs

The umap package [30] was used to analyze the sample heterogeneity between HT and normal thyroid tissues. The limma package [31] was used to screen DEGs between HT patients and controls in the GSE138198 dataset. |fold change (FC)| ≥ 1.5 and adjusted *p* < 0.05 were set as the threshold values of DEG identification. The ggplot2 package [32] and ComplexHeatmap package [33] were utilized to visualize the distribution of DEGs. Then, the extracellular protein gene list from the Uniprot database and the list from the HPA database were intersected with DEGs. The intersection of the above results was identified to be the EP-DEGs, and we analyzed the EP-DEGs between HT and normal thyroid tissue samples from GSE138198.

### 2.3. Function and Pathway Enrichment Analysis

Enrichment analysis has been widely used in recent years to identify gene properties, based on the hypothesis that genes with similar expression profiles may be regulated by common pathways and involved in related functions [34]. In this study, GO [35] and KEGG [36] enrichment analyses were performed by ClusterProfiler package [37] to find potential molecular mechanisms, for which adjusted *p* < 0.05 and count ≥ 2 were selected as cutoff criteria. The biological processes (BPs), cellular components (CCs), and molecular functions (MFs) of EP-DEGs were identified and the KEGG pathway enrichment analysis was performed on upregulated EP-DEGs and downregulated EP-DEGs, respectively. Dot plot, bar plot, and circle graph were used to visualize enrichment analysis results.

### 2.4. Construction of PPI Network

The STRING online database (http://string-db.org, accessed date: 10 September 2023) was utilized to identify the PPI network in the functional protein association network using EP-DEGs as the query proteins [38]. To detect hub EP-DEGs, we used the Molecular Complex Detection (MCODE) [39] and CytoHubba plugins in Cytoscape for module analysis, setting the parameters according to degree methods. The maximal clique centrality (MCC) method in CytoHubba [40] was used to predict the essential proteins from the PPI network, and the top eight hub genes were identified as the hub EP-DEGs.

### 2.5. Evaluation and Validation of the Key EP-DEGs

The limma package was used to confirm the expression differences of the hub EP-DEGs between HT patients and controls in the GSE29315 dataset. |log2 fold change (FC)| ≥ 1 and adjusted *p* < 0.05 were set as the cutoff criteria, and the hub EP-DEGs with significant differences were considered the key EP-DEGs. The ROC curve analysis was performed using the pROC package [41] to evaluate the diagnostic value of the hub EP-DEGs in GSE29315.

### 2.6. Immune Infiltration Analysis

The single-sample gene set enrichment analysis (ssGSEA) algorithm is a method for characterizing the cell composition of complex tissues from their gene expression profiles [42]. In this study, we utilized ssGSEA to examine the distribution of immune cell subtypes in both HT and normal control (NC) samples from GSE29315. We then compared the levels of different immune cells between the HT and the NC samples using the Wilcoxon rank sum test. To evaluate the correlation between the key EP-DEGs and immune cells in HT, Spearman correlation analysis was also performed.

### 2.7. Validation of Clinical Samples

To exclude the interference of thyroid carcinoma, we collected thyroid tissues from patients who underwent thyroidectomy due to benign thyroid nodules. HT was diagnosed according to surgical pathology. Normal thyroid samples were defined as thyroid tissues without HT or other diffuse thyroid lesions. The HT and normal thyroid tissues sited at least 1 cm away from the nodules were obtained and used in this study. A total of 6 HT thyroid samples and 5 normal thyroid samples were collected in Union Hospital, Tongji Medical College, Huazhong University of Science and Technology (Wuhan, China) from July 2023 to August 2023. The ethical approval number of the Ethics Committee of the Union Hospital is 2017(S062), and all subjects gave informed consent. The complete RNA of thyroid specimens was extracted and then reverse-transcribed into cDNA. RT-PCR was performed, and the relative levels of different key EP-DEGs were compared between HT and normal samples. The H&E and IHC staining of several key EP-DEGs were also performed to further validate the expression levels in clinical samples. Antibodies against CCL5, GZMK, CXCL9, CXCL10, CXCL11, and CXCL13 were all purchased from Proteintech (Wuhan, China).

### 2.8. RNA Extraction and RT-PCR

Total RNA of the HT and normal thyroid tissues was extracted using the Trizol (Vazyme, Nanjing, China) extraction method, and reverse transcription was conducted using HiScript III qRT SuperMix (Vazyme). PCR was conducted using the 2x Taq SYBR master mix purchased from Vazyme under the manufacturer’s instructions. Each sample underwent a RT-PCR assay using a BioRad CFX96 Real-Time PCR Detection System (Carlsbad, CA, USA) in triplicate. Relative gene expressions of key EP-DEGs at the RNA level were calculated by the ΔΔCt method, and GAPDH served as the internal control. Primers used in our study are listed as follows: GAPDH: forward primer: GGAGCCAAAAGGGTCATCACTC, reverse primer: GAGGGGCCATCCACAGTCTTCT; CCL5: forward primer: GAGTATTTCTACACCAGTGGCAAG, reverse primer: TCCCGAACCCATTTCTTCTCT; GZMK: forward primer: AGAAGTCATGTTACTGTCCTAAGTCG, reverse primer: TTGTAACTTAATTTGTATGAGGCGGGAC; CXCL9: forward primer: GAGTGCAAGGAACCCCAGTA, reverse primer: TTTCTCGCAGGAAGGGCTTG; CXCL10: forward primer: GAGCCTACAGCAGAGGAACC, reverse primer: GAGAGGTACTCCTTGAATGCCA; CXCL11: forward primer: GACGCTGTCTTTGCATAGGC, reverse primer: GGATTTAGGCATCGTTGTCCTTT; CXCL13: forward primer: AGGCCACGGTATTCTGGAAG, reverse primer: AGCTTGGGGAGTTGAAGACA.

### 2.9. H&E Staining and IHC Staining

H&E staining was performed to observe the histopathological features of the HT and normal thyroid tissues, while IHC staining was carried out to assess the protein expression levels of key EP-DEGs. For H&E and IHC staining, tissues were fixed in 4% paraformaldehyde and embedded with paraffin. Sections were prepared and dewaxed, and H&E staining was subsequently conducted. For IHC staining, antigens were unmasked with citrate buffer and 3% H_2_O_2_ was used to avoid the influence of endogenous peroxidases. Antibodies against CCL5, GZMK, CXCL9, CXCL10, CXCL11, and CXCL13 (Proteintech) were used to capture the expression of key EP-DEGs. Sections were visualized by a microscope (Olympus, Tokyo, Japan).

### 2.10. Statistical Methods

SPSS 24.0 (SPSSInc., Chicago, IL, USA) and R 4.3.1 (Comprehensive R Archive Network) were used for statistical analysis, and the R packages and statistical methods used in this study are presented above. Differences were considered statistically significant at * *p* < 0.05, ** *p* < 0.01, and *** *p* < 0.001.

## 3. Results

### 3.1. Identification of DEGs

Bulk gene expression data were extracted from the dataset GSE138198, and the sample-to-sample heterogeneity was detected by principal component analysis (PCA) [43]. As shown in Figure 2A, there were significant differences in gene expression between HT and normal thyroid samples. A total of 203 DEGs between these two groups were screened (Figure 2B), among which the top 20 upregulated DEGs and top 20 downregulated DEGs were exhibited in the heat map (Figure 2C).

### 3.2. Identification of EP-DEGs

To identify the genes encoding extracellular proteins that are differentially expressed between HT and normal thyroid samples, we referred to the annotated extracellular protein genes from two public protein annotation databases. A total of 4276 extracellular protein genes from the Uniprot database and 1903 genes from the HPA database were intersected with DEGs, and 19 EP-DEGs were screened out (Figure 3A). Among the 19 EP-DEGs, 16 were upregulated and 3 were downregulated (Figure 3B). The EP-DEGs and relative expression levels are exhibited in the heat map (Figure 3C).

### 3.3. Function and Pathway Enrichment Analysis

GO and KEGG enrichment analyses were performed to investigate the potentially involved functions and pathways of EP-DEGs. EP-DEGs were mainly enriched in the antimicrobial humoral response and chemokine-mediated signaling pathway of BPs, chemokine receptor binding and chemokine activity of MFs, and chemokine signaling pathway and viral protein interaction with cytokine and the cytokine receptor of KEGG (Figure 4A). The upregulated EP-DEGs and downregulated EP-DEGs were analyzed in the KEGG pathway, respectively. The upregulated genes are enriched in the viral protein interaction with cytokine and cytokine receptor, the chemokine signaling pathway, the toll-like receptor signaling pathway, and cytokine–cytokine receptor interaction (Figure 4B). Downregulated genes are enriched in parathyroid hormone synthesis, secretion and action, rheumatoid arthritis, and endocrine and other factor-regulated calcium reabsorption (Figure 4C). The cnetplot function in the ClusterProfiler package was used to display the genes enriched in the top five processes with the most significant *p*-value of BPs (Figure 4D), CCs (Figure 4E), and MFs (Figure 4F).

### 3.4. Establishment of PPI Network and Identification of Hub EP-DEGs

To explore the interaction between the proteins corresponding to EP-DEGs, the STRING database was used to construct a PPI network, which included 15 nodes and 36 edges. As shown in Figure 5A, the more the color and the wider the edge, the stronger the evidence for the interaction between proteins (details in Appendix A). Using the MCODE plugin in Cytoscape, eight genes in module 1 were identified as potential hub genes, and they were XCL1, SAA2, CCL5, CXCL9, CXCL10, CXCL11, CXCL13, and GZMK (Figure 5B). The MCC method of the CytoHubba plugin in Cytoscape was also used to screen the top 10 hub genes with the highest scores (Figure 5C). Combining the results of the MCODE plugin and the CytoHubba plugin, we determined XCL1, SAA2, CCL5, CXCL9, CXCL10, CXCL11, CXCL13, and GZMK as the hub EP-DEGs for further exploration and validation.

### 3.5. Evaluation and Validation of Key EP-DEGs

The dataset GSE29315 was used to verify the candidate hub EP-DEGs. The expression levels of the hub genes were analyzed again and six of them were significantly upregulated in HT samples (Figure 6A). CCL5, CXCL9, CXCL10, CXCL11, CXCL13, and GZMK were identified to be the key EP-DEGs. ROC curves were plotted to investigate the diagnostic value of the key EP-DEGs, and it was observed that the AUC values of all key EP-DEGs were above 0.95 (Figure 6B–G).

### 3.6. Immune Infiltration Analysis

ssGSEA was utilized to assess the distribution of immune cell subtypes in the tissues from the GSE29315 dataset. Figure 7A demonstrates significant differences between the HT group and the normal control (NC) group in various immune cell types, including CD8+ T cells, DCs, mast cells, plasmacytoid dendritic cells (pDCs), and Th2 cells. As previously reported, CD8+ T cells, DCs, and Th2 cells might contribute to thyrocyte destruction and prompt the pathogenesis of HT [18,24]. We therefore calculated the correlations between the expressions of the key EP-DEGs and the proportion of differentially infiltrated immune cells. Our results demonstrated that CXCL11 was most correlated with CD8+ T cells (R = 0.587, Figure 7B), and all the key EP-DEGs have correlations with both DCs (Figure 7C) and Th2 cells (Figure 7D).

### 3.7. Preliminary Validation of Clinical Samples

To further validate the above bioinformatics findings, six HT thyroid samples and five normal thyroid samples were collected in Union Hospital, Tongji Medical College, Huazhong University of Science and Technology (Wuhan, China). Patients’ medical history and laboratory parameters were collected (details in Appendix A). The expression levels of all key EP-DEGs in HT samples were significantly higher than those in normal thyroid samples, both at the RNA levels (Figure 8A) and protein levels (Figure 8B). As shown in the representative histology of the clinical thyroid samples, histopathologic features of HT include lymphoplasmacytic infiltration, lymphoid follicle formation, and parenchymal atrophy, and the high levels of key EP-DEGs were mainly distributed around infiltrating lymphocytes and lymphoid follicles (Figure 8B).

## 4. Discussion

In this study, we explored the gene expression profiles of thyroid samples and screened out different extracellular proteins between HT and controls using bioinformatics methods. A total of 19 EP-DEGs were screened by the GSE138198 dataset and the main biological processes and pathways they participated in were the chemokine signaling pathway and viral protein interaction with cytokine and cytokine receptor. The GSE29315 dataset was subsequently used to verify the key EP-DEGs, which are CCL5, GZMK, CXCL9, CXCL10, CXCL11, and CXCL13. Immune infiltration analysis indicated that the key EP-DEGs were significantly positively correlated with CD8+ T cells, DCs, and Th2 cells, which might prompt the pathogenesis of HT [18,24,25,26]. Additionally, clinical samples were collected for further validation, and the key extracellular proteins were found to be enriched in the infiltrating lymphocytes and lymphoid follicles of HT thyroid tissues. These results of extracellular proteins may improve the comprehension of pathogenic mechanisms, diagnostic protocols, and therapeutic strategies of HT.

Among the six key EP-DEGs, CCL5, CXCL9, CXCL10, CXCL11, and CXCL13 are all chemokines mainly secreted by monocytes, endothelial cells, fibroblasts, mesenchymal lymphoid tissue organizer cells, DCs, and T follicular helper cells. Proteins encoded by these key EP-DEGs could induce chemotaxis, promote differentiation of immune cells, and cause tissue extravasation [44,45,46]. As for GZMK, it is a member of the serine proteases secreted by granules inside cytotoxic cells of the immune system and performs multiple intracellular and extracellular functions including cytotoxicity, endothelial activation, pro-inflammatory cytokine response, and pro-apoptosis [47]. These key extracellular proteins could be secreted into the extracellular matrix by specific cells in the pathological process of HT and could act as ligands to bind to receptors on specific cells, hence transmitting signals for cell-to-cell communication and mediating processes such as apoptosis.

Because the origin and development of HT involve many immune molecules and signaling pathways, it is difficult to completely understand the pathogenesis of HT. CCL5 was found to be remarkably elevated in tumor tissues collected from PTC patients coexistent with HT than those without HT [48], and CXCL9, CXCL10, CXCL11, and CXCL13 were also reportedly upregulated in thyroid tissues from HT patients [26,49]. Our study was consistent with the previous literature reports on the differential expression levels and further revealed the correlation between these chemokines and other immune cells. GZMK has not been reported in thyroid diseases so far, and our study illustrated its increased expression level and its correlation with some immune cells involved in the disease progression of HT. Our results indicated that the key extracellular proteins in HT might mainly composed of some chemokines and serine protease GZMK, which were primarily enriched in the infiltrating lymphocytes and lymphoid follicles of thyroid tissues of HT patients. Based on the property that extracellular proteins can be secreted and detected in body fluids, the key EP-DEGs, along with TPOAb and TGAb, could serve as biomarkers of HT individually or in combination, as well as being potential therapeutic targets due to the pathways and immune cells they participate in.

However, some limitations exist in the present study. Firstly, though we used clinical specimens for validation, the main analyses were based on high-throughput sequencing data and bioinformatics. Secondly, the sample size was relatively small, and larger datasets may help to further verify the present results. In addition, the study lacked detailed experiments to verify the specific mechanisms of the key EP-DEGs in HT. Finally and most importantly, extracellular proteins could be detected in clinical tissues and also some body fluids, but due to the limitation of the source of specimens, our study was based only on thyroid tissues and no other body fluids such as blood. Considering the potential bias, it is important to conduct further biological and clinical experiments in our subsequent work. If these factors are validated in the future and translated into clinical practice, they may be used as potential diagnosis biomarkers and therapeutic strategies for HT.

## Figures and Tables

**Figure 1 biomedicines-11-03127-f001:**
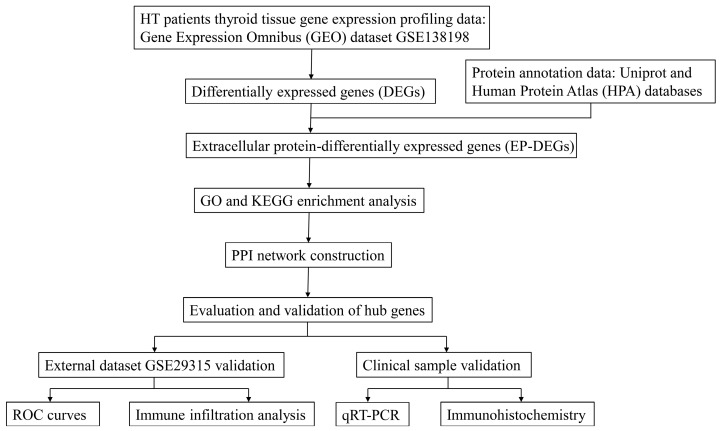
Flowchart of the study.

**Figure 2 biomedicines-11-03127-f002:**
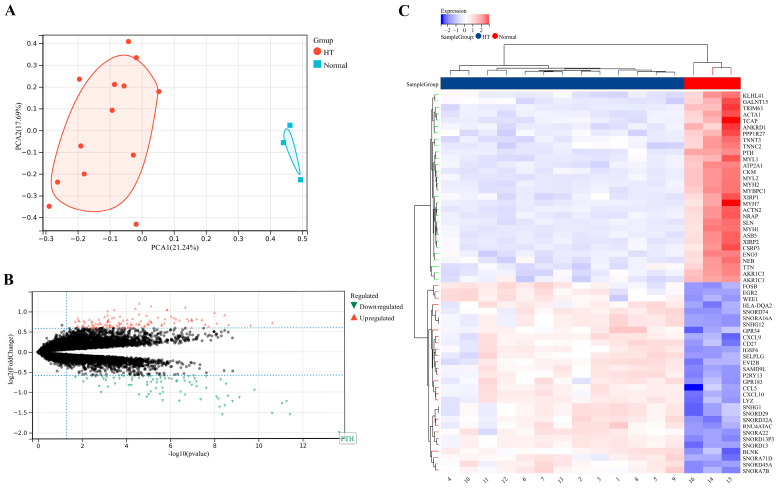
Identification of DEGs. (**A**) PCA plot. The center points of the HT group and the normal thyroid group are far apart in space. (**B**) Volcano map of all DEGs in the HT group and the normal thyroid group. (**C**) Heatmap of top 20 upregulated DEGs and top 20 downregulated in the HT group and the normal thyroid group.

**Figure 3 biomedicines-11-03127-f003:**
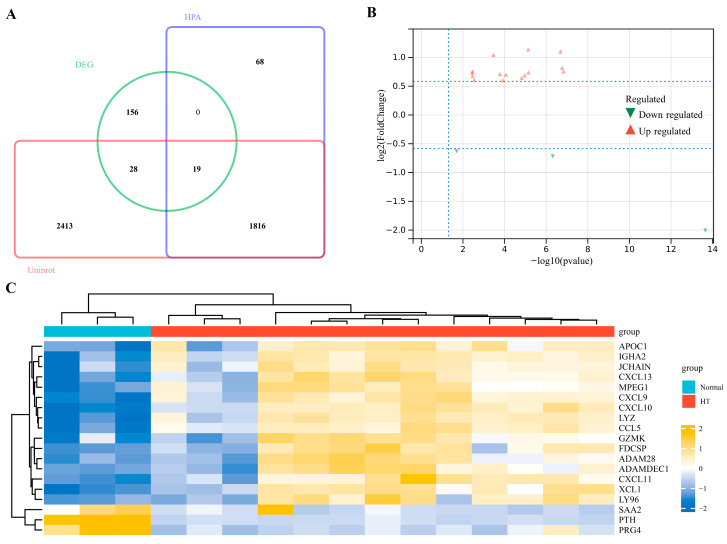
Identification of EP-DEGs. (**A**) Venn diagram of genes among DEGs and extracellular protein genes from HPA and Uniprot. (**B**) Volcano map of EP-DEGs in the HT group and the normal thyroid group. (**C**) Heatmap of EP-DEGs in the HT group and the normal thyroid group.

**Figure 4 biomedicines-11-03127-f004:**
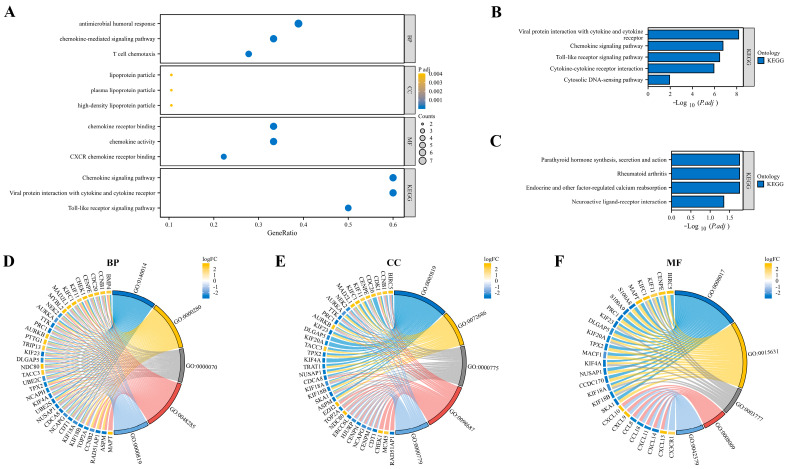
Function and pathway enrichment analysis. (**A**) GO and KEGG enrichment of EP-DEGs. The dots show the top 3 processes enriched by EP-DEGs. (**B**) Pathways to which the upregulated EP-DEGs are enriched. (**C**) Pathways to which the downregulated EP-DEGs are enriched. (**D**–**F**) Circle graphs in GO enrichment of EP-DEGs. Graphs show the EP-DEGs enriched in the top 5 GO categories of BPs, CCs, and MFs, respectively.

**Figure 5 biomedicines-11-03127-f005:**
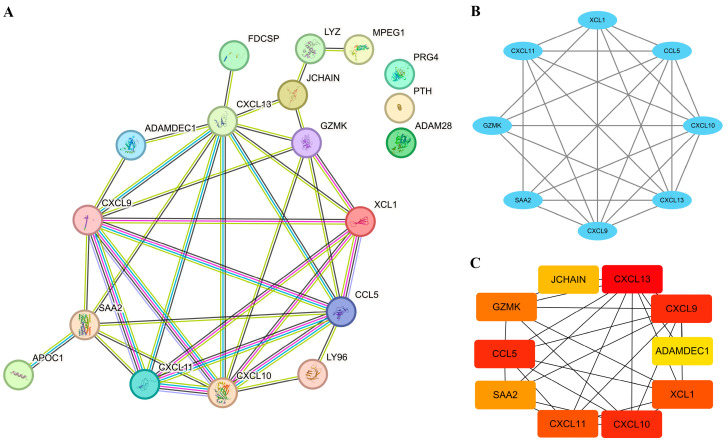
Establishment of PPI network and identification of hub EP-DEGs. (**A**) The STRING database is used to construct the PPI network of EP-DEGs. (**B**) The node gene cluster with the highest score constructed by the MCODE plugin in Cytoscape consists of 8 EP-DEGs. (**C**) The figure shows the top 10 hub EP-DEGs constructed by the MCC method using the Cytohubba plugin in Cytoscape. The redder the gene, the higher the score.

**Figure 6 biomedicines-11-03127-f006:**
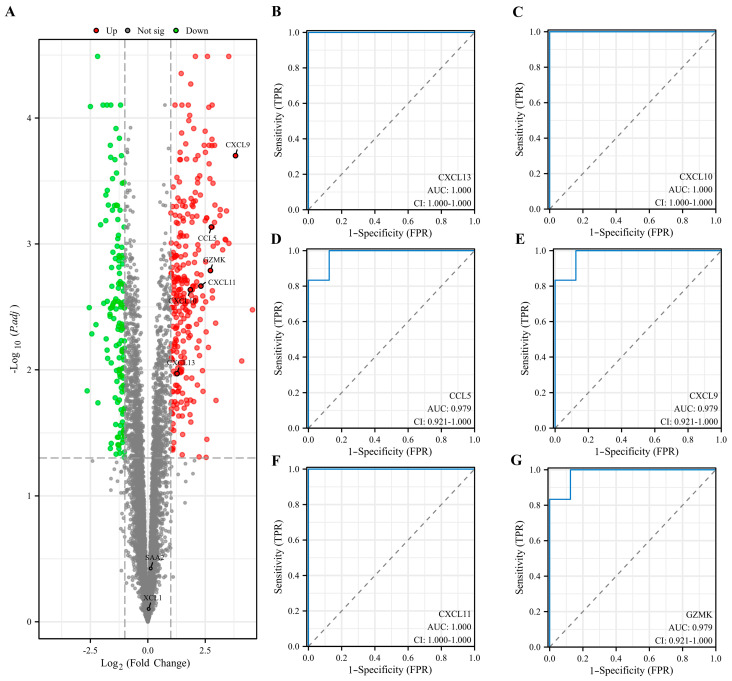
Evaluation and validation of key EP-DEGs in the GSE29315 dataset. (**A**) Volcano map of all DEGs in the HT group and the normal thyroid group in the GSE29315 dataset. Hub EP-DEGs in the GSE138198 are labeled. (**B**–**G**) ROC curves depicting the diagnostic efficacy of 6 key EP-DEGs in the GSE29315 dataset, respectively.

**Figure 7 biomedicines-11-03127-f007:**
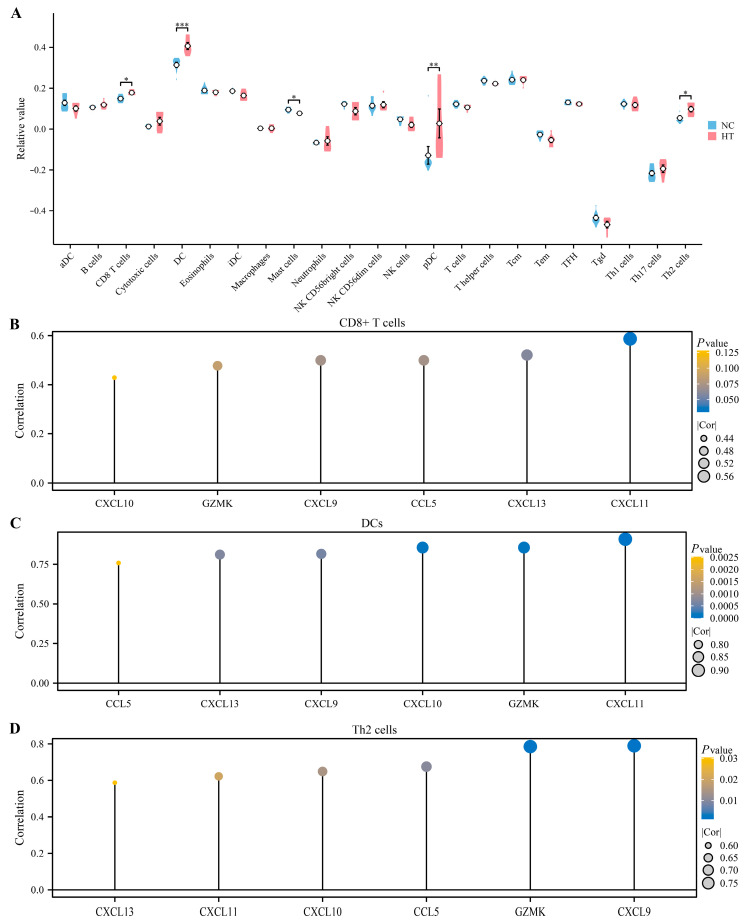
Immune infiltration analysis. (**A**) The violin plot exhibits the expression levels of immune cells in the HT group and the normal control (NC) group analyzed by the ssGSEA algorithm. (**B**–**D**) Correlation of the key EP-DEGs with immune cells. Differences were considered statistically significant at * *p* < 0.05, ** *p* < 0.01, and *** *p* < 0.001.

**Figure 8 biomedicines-11-03127-f008:**
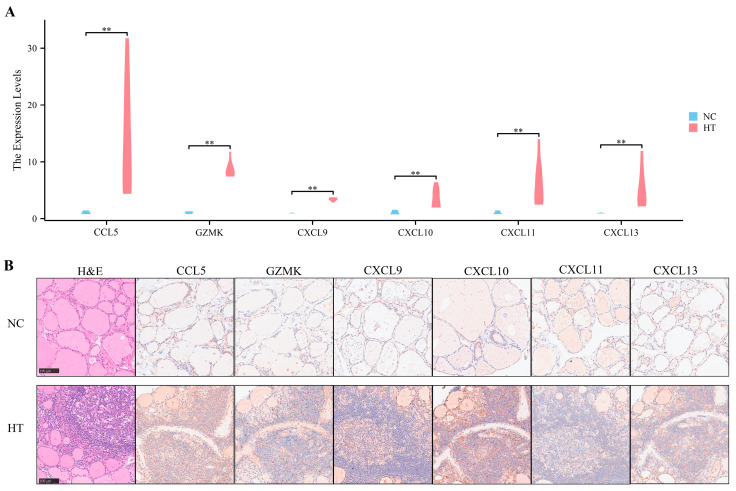
Preliminary validation of clinical samples. (**A**) The violin plot exhibits the RNA levels of the key EP-DEGs in the thyroid tissues of the HT group and normal control (NC) group. (**B**) Representative histology of the thyroid tissues of the HT group and NC group by H&E and IHC staining. Differences were considered statistically significant at ** *p* < 0.01.

## Data Availability

The original contributions presented in the study are included in the article. Some public data are from the GEO database.

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
