# Peer review of "Identification and Preliminary Clinical Validation of Key Extracellular Proteins as the Potential Biomarkers in Hashimoto’s Thyroiditis by Comprehensive Analysis"

_biomedicines, 2023, doi:10.3390/biomedicines11123127_

Round 1

Reviewer 1 Report

Comments and Suggestions for Authors

Overall, the authors apply multiple techniques to analyse extracellular proteins as the potential biomarkers in Hashimoto’s thyroiditis. Methods have been applied in a wide range, and there is not doubt about the methods.

However, there are several critical points, that have also been mentioned in the discussion and need to be resolved before publication:

1. sample size is too small

2. verification on new patient material is required. 

For clinical markers, both points are essential and required before acceptance can be granted.

Besides that, epidemiological terms such as prevalence (lines 38, 41) need to be explained more clearly, and not just cited. 

Author Response

Dear reviewer,

We feel great thanks for your professional comments concerning our manuscript “Identification and clinical validation of key extracellular proteins as the potential biomarkers in Hashimoto’s thyroiditis by comprehensive analysis”. Those comments are all valuable and very helpful for further improving our paper. We read the comments carefully and have made corresponding corrections which we hope to meet with your approval.

Yours sincerely,

Zihan Xi, Tinglin Yang, Tao Huang, Jun Zhou, and Peng Yang

The main corrections in the paper and the response to your comments are as follows. All modifications in the manuscript have been marked up by using the “track changes” function in MS Word.

Comment #1: Overall, the authors apply multiple techniques to analyse extracellular proteins as the potential biomarkers in Hashimoto’s thyroiditis. Methods have been applied in a wide range, and there is not doubt about the methods.

However, there are several critical points, that have also been mentioned in the discussion and need to be resolved before publication:

  1. sample size is too small

Response #1: We apologize for the small sample size due to the accessibility of HT and normal thyroid tissues. To the reduce bias resulting from the small sample size, we detected the expressions of key EP-DEGs in clinical samples at both RNA levels and protein levels. Significantly higher expressions of key EP-DEGs were detected in HT samples, which can preliminarily support our conclusions.

Comment #2: 2. verification on new patient material is required. For clinical markers, both points are essential and required before acceptance can be granted.

Response #2: As secretory proteins with potential clinical application prospects, collecting the patients’ body fluids including blood to detect the levels of EP-DEGs is indeed a reliable verification method. However, due to the limitation of the source of specimens, our study only detected EP-DEGs in thyroid tissue. We reflected on this issue in section 4. Discussion (Line 473-479, Page 12). In future work, we are planning to collect more specimens from both HT patients and healthy people for further exploration and validation.

Comment #3: Besides that, epidemiological terms such as prevalence (lines 38, 41) need to be explained more clearly, and not just cited.

Response #3: Detailed epidemiological terms, as well as the etiology and pathogenesis of HT are introduced in section 1. Introduction (Line 39, Page 1; and Line 67, Page 2).

Reviewer 2 Report

Comments and Suggestions for Authors

Review for the article “Identification and clinical validation of key extracellular proteins as the potential biomarkers in Hashimoto’s thyroiditis by 3 comprehensive analysis”.

The authors tried to identify extracellular biomarkers using Bioinformatics analysis in order to perform better pathogenic mechanisms, diagnostic and therapeutic strategies of Hashimoto thyroiditis (HT). They also performed RT-PCR analysis and confirmed that expression levels of all key EP-DEGs in HT samples were significantly higher than those in normal thyroid samples.

 Comments

The objectives and the research design were appropriate.

Bioinformatics analysis are well described. But, please indicate the details for RT-PCR method (reagents, equipment, protocols) performed in order to validate clinical samples.

The results are clearly presented. The conclusions supported by the results. 

Author Response

Dear reviewer,

We feel great thanks for your professional comments concerning our manuscript “Identification and clinical validation of key extracellular proteins as the potential biomarkers in Hashimoto’s thyroiditis by comprehensive analysis”. Those comments are all valuable and very helpful for further improving our paper. We read the comments carefully and have made corresponding corrections which we hope to meet with your approval.

Yours sincerely,

Zihan Xi, Tinglin Yang, Tao Huang, Jun Zhou, and Peng Yang

The main corrections in the paper and the response to your comments are as follows. All modifications in the manuscript have been marked up by using the “track changes” function in MS Word.

Comment: Review for the article “Identification and clinical validation of key extracellular proteins as the potential biomarkers in Hashimoto’s thyroiditis by 3 comprehensive analysis”.

The authors tried to identify extracellular biomarkers using Bioinformatics analysis in order to perform better pathogenic mechanisms, diagnostic and therapeutic strategies of Hashimoto thyroiditis (HT). They also performed RT-PCR analysis and confirmed that expression levels of all key EP-DEGs in HT samples were significantly higher than those in normal thyroid samples.

Comments:

The objectives and the research design were appropriate.

Bioinformatics analysis are well described. But, please indicate the details for RT-PCR method (reagents, equipment, protocols) performed in order to validate clinical samples.

The results are clearly presented. The conclusions supported by the results.

Response: Thank you for your comments. We have added the details for RT-PCR assays in section 2. Materials and Methods (Line 273-291, Page 5). Besides, the methods for H&E and IHC staining were also described (Line 292-302, Page 5).

Reviewer 3 Report

Comments and Suggestions for Authors

The paper describes a nice study focused on the identification of extracellular proteins overexpressed in the Hashimoto’s thyroiditis (HT) . The authors have demonstrated their expertise in practical application of bioinformatics to reach their goal in an elegant way. The study is well designed and well described. As a reviewer I have some remarks, though:

1.       The introduction consists of 4 paragraphs and the third one makes up nearly a half of the section. However, that paragraph is actually – except for the first sentence - a brief description of the study design and as such should be moved to the Material and methods section – I would suggest combining it with the subsection 2.1. Study design and data acquisition

2.       The rest of the introduction lacks a deeper insight into the current state of knowledge about the pathogenesis of HT. It is really hard to touch that subject and not to mention the iodine exposure. I am not sure whether this omission was purposeful but it comes in line with the fact that all identified extracellular proteins are involved in the inflammatory process that occurs as a conclusion of the HT pathogenesis. They don’t give us any clue about what triggers the inflammation. And we can safely assume that these proteins are not specific to the thyroid gland or HT. We should expect to find their overexpression wherever the lymphocytic infiltration occurs. Anyway, the authors should better elaborate on the pathogenesis of HT in the introduction. And they should be a bit more modest when they write about their study as if it addressed some key questions about  the pathogenesis of HT.

3.       The authors are prudent to mention the lack of verification of the clinical usefulness of the identified proteins as biomarkers of HT by determination of their concentration in the blood of patients with HT and healthy controls (when enumerating limitations to their study). I agree, this is an important limitation and that is why I object using the phrase ‘and clinical validation’ in the title as it is simply misleading. Because the identified proteins are not specific to the thyroid I doubt that they could serve as biomarkers of HT as such, but they might be potentially useful (at least one or two of them) as indicators of the intensity of the inflammation – more accurate than TPOAb or TGAg. This issue could be better discussed.

Comments on the Quality of English Language

I'm not a native speaker but I found the manuscript written in a very conprehendible way with only a few places where I was supprised by a strange selection of the words. That should be easy to correct by the editor.

Author Response

Dear reviewer,

We feel great thanks for your professional comments concerning our manuscript “Identification and clinical validation of key extracellular proteins as the potential biomarkers in Hashimoto’s thyroiditis by comprehensive analysis”. Those comments are all valuable and very helpful for further improving our paper. We read the comments carefully and have made corresponding corrections which we hope to meet with your approval.

Yours sincerely,

Zihan Xi, Tinglin Yang, Tao Huang, Jun Zhou, and Peng Yang

The main corrections in the paper and the response to your comments are as follows. All modifications in the manuscript have been marked up by using the “track changes” function in MS Word.

Comment #1: The paper describes a nice study focused on the identification of extracellular proteins overexpressed in the Hashimoto’s thyroiditis (HT) . The authors have demonstrated their expertise in practical application of bioinformatics to reach their goal in an elegant way. The study is well designed and well described. As a reviewer I have some remarks, though:

  1. The introduction consists of 4 paragraphs and the third one makes up nearly a half of the section. However, that paragraph is actually – except for the first sentence - a brief description of the study design and as such should be moved to the Material and methods section – I would suggest combining it with the subsection 2.1. Study design and data acquisition

Response #1: Thank you for your suggestion. We combined the description of our study with subsection 2.1 (Line 164-184, Page 3), which indeed helped to improve the logic of our manuscript.

Comment #2: The rest of the introduction lacks a deeper insight into the current state of knowledge about the pathogenesis of HT. It is really hard to touch that subject and not to mention the iodine exposure. I am not sure whether this omission was purposeful but it comes in line with the fact that all identified extracellular proteins are involved in the inflammatory process that occurs as a conclusion of the HT pathogenesis. They don’t give us any clue about what triggers the inflammation. And we can safely assume that these proteins are not specific to the thyroid gland or HT. We should expect to find their overexpression wherever the lymphocytic infiltration occurs. Anyway, the authors should better elaborate on the pathogenesis of HT in the introduction. And they should be a bit more modest when they write about their study as if it addressed some key questions about the pathogenesis of HT.

Response #2: The pathogenesis of HT can be concluded as a result of losing immune tolerance, and a brief description of specific pathogenesis was added in section 1. Introduction (Line 56-62, Page 2). Besides, the epidemiology and etiology of HT including iodine exposure were also reviewed in the Introduction section (Line 38-55, Page 1-2). These proteins are not specific to the thyroid gland or HT, and they're very common in the human body. In our study, we found their overexpression in HT tissues, especially wherever the lymphocytic infiltration occurs by IHC staining. In addition, we would like to apologize for our overconfident statements in the paper, and certain revisions have been made.

Comment #3: The authors are prudent to mention the lack of verification of the clinical usefulness of the identified proteins as biomarkers of HT by determination of their concentration in the blood of patients with HT and healthy controls (when enumerating limitations to their study). I agree, this is an important limitation and that is why I object using the phrase ‘and clinical validation’ in the title as it is simply misleading. Because the identified proteins are not specific to the thyroid I doubt that they could serve as biomarkers of HT as such, but they might be potentially useful (at least one or two of them) as indicators of the intensity of the inflammation – more accurate than TPOAb or TGAg. This issue could be better discussed.

Response #3: Since the HT and normal thyroid tissues were all from clinical patients, we used the phrase ‘clinical validation’, and we are willing to change this expression to "preliminary clinical validation" in the title and text of our manuscript to make it more precise (Line 2, Page1; Line 405, Page 10; Line 418, Page 11). We are applying for new ethical approval to further validate the findings of this paper in the blood of patients and to further explore the relationship between the EP-DEGs and the inflammatory level and development of HT.

Comment #4: Comments on the Quality of English Language:

I'm not a native speaker but I found the manuscript written in a very conprehendible way with only a few places where I was supprised by a strange selection of the words. That should be easy to correct by the editor.

Response #4: Thank you for your comments. We have revised part of the language of our manuscript. We also sincerely hope that you and the editor can point out our specific language problems.

Round 2

Reviewer 2 Report

Comments and Suggestions for Authors

Accept in the present form. The authors described in details the RT-PCR methods. 

Reviewer 3 Report

Comments and Suggestions for Authors

The authors have addressed all my concerns and I don't have any other remarks. The term 'preliminary clinical validation' proposed in the revision is a good compromise in my opinion. The introduction has gained the necessary balance. I have to admit that I couldn't find the words that struck me as strangely selected in the original submission.